# Bone Involvement in Systemic Lupus Erythematosus

**DOI:** 10.3390/ijms23105804

**Published:** 2022-05-22

**Authors:** Valeria Rella, Cinzia Rotondo, Alberto Altomare, Francesco Paolo Cantatore, Addolorata Corrado

**Affiliations:** Rheumatology Clinic, Department of Medical and Surgical Sciences, University of Foggia, 71122 Foggia, Italy; cinzia.rotondo@gmail.com (C.R.); alto.albe@gmail.com (A.A.); francescopaolo.cantatore@unifg.it (F.P.C.)

**Keywords:** systemic lupus erythematosus, osteoporosis, avascular necrosis, osteomyelitis

## Abstract

Systemic lupus erythematosus (SLE) is a chronic autoimmune disease characterized by a wide variability of clinical manifestations due to the potential involvement of several tissues and internal organs, with a relapsing and remitting course. Dysregulation of innate and adaptive immune systems, due to genetic, hormonal and environmental factors, may be responsible for a broad spectrum of clinical manifestations, affecting quality of life, morbidity and mortality. Bone involvement represents one of the most common cause of morbidity and disability in SLE. Particularly, an increased incidence of osteoporosis, avascular necrosis of bone and osteomyelitis has been observed in SLE patients compared to the general population. Moreover, due to the improvement in diagnosis and therapy, the survival of SLE patient has improved, increasing long-term morbidities, including osteoporosis and related fractures. This review aims to highlight bone manifestations in SLE patients, deepening underlying etiopathogenetic mechanisms, diagnostic tools and available treatment.

## 1. Introduction

Systemic lupus erythematosus (SLE) is a chronic and clinically heterogeneous disease, with a not-fully understood aetiology, autoimmune pathogenesis and a relapsing and remitting course. Incidence and prevalence rates vary between countries due to differences in patient characteristics, particularly genetic background, ethnic group, geographic region, socioeconomic status and environmental exposures, with the highest rates observed in North America (1.0—23.2 per 100,000 person-years and 15.3—241.0 per 100,000, respectively), in particular in African descendants, and in women of childbearing age (incidence ratio 8.0—15.1:1) [1]; probably, in a genetic predisposing substrate, hormonal and environmental factors’ interplay can trigger an autoimmune process. The innate and adaptive immune systems are involved through dysregulation of several cytokines, including type I interferons, complement activation, impaired clearance of nucleic acids after cell death, production of autoantibodies and immune complexes, in order to generate a self-sustained autoimmune process [2,3].

SLE may affect several organs with a heterogeneous and systemic involvement, whose clinical manifestations range from mucous and cutaneous lesions to severe, life-threatening organ damage [2,3,4]. In fact, diagnosis is often difficult or delayed and relies on physician’s expertise to combine clinical and immunological findings [3]. Although the diagnosis of SLE is clinical, classification criteria have been developed, also used to identify patient subsets for controlled trials [5,6]. Along with the clinical manifestations, comorbidities substantially contribute to the disease burden and mortality, in particular cardiovascular complications and increased infection risk [3,7].

SLE treatment is directed at the suppression of disease activity and the prevention of irreversible organ damage, leading to survival rates between 85% to 90% during the first 10 years [8], but iatrogenic damage is a major clinical issue in SLE, mostly related to long-term use of glucocorticoids (GCs) [9].

Bone involvement represents one of the most common causes of morbidity and disability in SLE. In particular, an increased incidence of osteoporosis, avascular osteonecrosis and osteomyelitis has been observed in SLE patients. This review aims to highlight these manifestations in SLE patients. For this purpose, the authors searched the Pubmed database for any published studies from 1999 to February 2022 using as keywords the following: “lupus erythematosus systemic”, “rheumatic diseases”, “bone”, “osteoimmunology”, “osteoporosis”, “fracture”, “osteonecrosis”, “avascular necrosis”, “osteomyelitis”, “glucocorticoid”, “risk factors”, “imaging”. The Pubmed searches were performed from June 2021 to February 2022.

## 2. SLE and Osteoporosis

Osteoporosis (OP) is a skeletal disease characterized by reduced bone mass and micro-architectural impairment of bone tissue, leading to increase bone fragility and fracture risk [10,11]. OP and related fragility fractures represent a major public health problem due to their impact on mortality and morbidity deriving from an increase in functional impairment and decreased quality of life [11]. Diagnosis of OP relies on bone mineral density (BMD) measurement using dual-energy X-ray absorptiometry (DXA) scanning or on vertebral or hip fracture in the absence of major trauma [12,13].

Several autoimmune rheumatic diseases may be complicated by systemic and local bone loss [14,15,16,17,18]. This process has a multifactorial physiopathology, encompassing treatment, immobilization and reduced physical activity due to musculoskeletal symptoms; further, systemic chronic inflammation plays a key role in the pathogenesis of bone loss [17]. Indeed, the inflammatory cytokines network leads to an uncoupling of bone resorption and formation and consequent bone loss and increased fracture risk [14,15,16,17,18]. A reduction in BMD is a frequent finding in SLE patients and an increased fragility fracture rate is observed in these patients, even with normal BMD values [19].

### 2.1. Epidemiology of Osteoporosis

In SLE patients, an increased prevalence of OP is observed; owing to improvement in diagnosis and therapy, the survival of SLE patients has improved but the long-term morbidities, including OP and related fractures, have become more frequent [14,19]. Osteopenia is reported in 24–74% and OP in 1.4–68.7% of SLE patients in cohort studies: these different frequencies are consequences of differences in size, age, gender, ethnic background, disease severity, medication use and study design between studies [16,20].

Along with traditional independent risk factors, disease-related factors affect the higher incidence and prevalence of OP and related fractures than general population [14,16], as shown in Table 1.

### 2.2. Fracture Risk

Osteoporotic fractures represent one of the major complications of SLE, reducing patients’ quality of life but also affecting healthcare and economic burden [25]. As reported by Kim et al., SLE patients showed a higher osteoporotic fracture rate than a control cohort (HR 2.964), even after adjustment for confounding variables; further, they reported that, in subgroup analysis, male SLE patients or SLE patients aged 40 to 65 years showed a higher osteoporotic fracture rate than women SLE patients or SLE patients aged ≥ 65 years, respectively (HR 4.706 vs. HR 2.899 between male SLE patients and female SLE patients; HR 3.383 vs. HR 2.223 between SLE patients aged between 40 and 65 years and SLE patients aged ≥ 65 years) [25].

The incidence of symptomatic osteoporotic fractures in SLE is increased by 1.2–4.7-fold versus sex-matched and age-matched healthy controls; nevertheless, fractures, particularly vertebral ones, may remain under-diagnosed because of a paucity of symptoms [16]. Vertebral fractures (VF) are the most common type of osteoporotic fracture and are associated with substantial morbidity, increased mortality and risk of future fractures; for these reasons, detecting VF represents an important issue [26]. A VF prevalence of 20 to 61% has been reported in SLE but 29 to 35.8% of patients with VF had a normal BMD, highlighting the limited value of BMD measurement and the role of other factors affecting bone quality in the assessment of fracture risk [26]. Moreover, a higher non-VF risk has been reported in SLE patients. In a recent large population-based study from Taiwan, a 3.2-fold higher incidence of hip fractures in SLE patients was reported, compared to age- and sex-matched controls; fracture risk was increased in younger subjects compared to the general population [27]. Assessment of vertebral deformities and measurement are advisable for SLE patients, since VFs increase the risk of further fractures and their detection is an indication to start anti-osteoporosis treatment even in osteopenic patients [16,19,28].

The finding that a high prevalence of morphometric VFs was present in SLE patients, while a third of these patients had normal BMD [29,30], supports the concept that BMD alone is insufficient for identifying high-risk patients, as it is not able to provide information about bone quality.

The assessment of bone quality and bone architecture is very difficult in current clinical practice, but, recently, a new non-invasive diagnostic tool has emerged as an important tool to assess bone architecture and bone quality.

The trabecular bone score (TBS) is a structural parameter that can be obtained from the textural greyscale analysis of DXA images and provide an indirect measurement of bone microarchitecture, irrespective of bone density. It has been shown that SLE patients have lower TBS values, which are associated with a higher risk of VFs even in patients with a T-score above the diagnostic threshold of OP [19,28,31] and that TBS in these patients has a higher predictive value than spine and hip BMD for VFs [19], suggesting an important role of TBS as an accurate and safe diagnostic tool for assessment of bone quality in chronic and systemic inflammatory rheumatic diseases, such as SLE, in order to quickly detect subjects at high risk of fragility fractures [19,28].

In addition to impaired bone mass and quality and previous osteoporotic fractures, other factors may affect fracture risk in SLE patients [16], as shown in Table 2. Notably, SLE patients with neurological complications, in particular stroke or seizures, have a greater risk for fractures, due to increased risk of falls, and adverse effects of antiepileptic drugs and anticoagulants on bone mass. The risk of falls might be also increased due to muscle weakness related to GCs use, inactivity and vitamin D deficiency, arthritis [16] and high prevalence of frailty in SLE patients, as reported by Kats et al. (according to Fried’s definition of frailty) [32].

### 2.3. Physiopathology of Bone Loss in SLE

Systemic inflammation affects both osteoclast and osteoblast function, leading to bone loss [16]. Osteoclasts are multinucleated bone resorbing cells which originate from the fusion of monocyte-macrophage precursor cells. Osteoclast differentiation, lifespan and activity are regulated by macrophage colony stimulating factor (M-CSF), receptor activator of nuclear factor-kB (RANK), its ligand RANKL and a decoy receptor osteoprotegerin (OPG) [15]. 

In SLE patients, there is an imbalance among different immune cell subsets, such as Th1/Th2 and Th17/regulatory T (Treg) cells with an abnormal expression of several proinflammatory cytokines, notably, IL-1, IL-6 and TNFα [18]. This cytokine network affects both osteoclast and osteoblast function, increasing the expression of RANKL by osteoblasts and osteocytes [15]. Further, TNFα promotes bone resorption indirectly (in conjunction with IL-6) by up-regulating RANKL expression, and directly, by promoting the differentiation of osteoclasts in synergy with RANKL [15,16]. Thus, in SLE patients, there is an increased production of RANKL and an RANKL/OPG imbalance leading to accelerated osteoclastogenesis [18]. Previous studies have observed that bone loss correlates with SLE disease activity and severity [15]. There is an association between organ damage and reduced BMD, as organ damage is greater in patients with prolonged active disease and in turn higher disease activity negatively affects BMD [14]. Another pro-inflammatory cytokine able to increase bone resorption is IL-17; an increased number of Th17 cells and elevated serum IL-17 levels are reported in SLE. IL-17 contributes to an imbalance in RANKL/OPG via the expression of RANKL in osteoblasts or activated T cells and could act in synergy with TNFα and other cytokines to influence osteoclast resorption [18]. Moreover, IL-21 produced by CD4+ T cells may enhance IL-6 secretion, amplifying its effects such as autoantibody production [18]. In addition, Treg cells are involved in impaired bone metabolism in SLE. In a physiologic state, Treg cells inhibit osteoclastogenesis through anti-inflammatory cytokines (such as IL-10) and CTLA4 signaling (a negative regulator of T-cell activation) [15]; it has been found that IL-6 produced by dendritic cells inhibits Treg cell function in mouse SLE models [18]. Furthermore, in SLE, high levels of oxidized low density lipoproteins (LDL) may induce T-cell activation, thereby increasing RANKL expression and TNFα production, and may also negatively affect bone formation by reducing osteoblast maturation [16,18]. TNFα also reduces bone formation by acting on the Wnt system [15]. TNFα suppresses osteoblast differentiation, by inhibiting the expression of insulin-like growth factor-1 (IGF-1), osterix, and runt-related transcription factor 2 (Runx2), and inducing the production of two potent inhibitors of Wnt signaling, Dickkopf-1 (DKK1) and sclerostin, that control both osteoclast and osteoblast differentiation [34]. As in several rheumatic diseases [35], also in SLE may an impaired osteoblast function be identified, with low serum levels of bone-formation markers such as osteocalcin related to disease activity [36,37,38]. In addition, the noteworthy role of interferon β (INFβ) in affecting bone formation has recently emerged. In addition to the well-known role of IFNα in SLE pathogenesis, which is produced by plasmacytoid dendritic cells and induces B, T, and myeloid dendritic cells activation, INFβ may inhibit bone marrow (BM) mesenchymal stem cells’ (MSCs) differentiation into mature osteoblasts. In SLE, INFβ is mainly produced by stromal tissue cells and BM-MSCs, based on a positive feedback loop. In SLE patients, a reduced expression of osteogenic markers (such as Runx2, bone sialoprotein, osteocalcin and alkaline phosphatase) in different stages of MSCs’ differentiation has been reported, consistent with defective osteogenesis. In SLE, INFβ may suppress osteogenesis by acting on the bone morphogenetic protein (BMP)/small mother against decapentaplegic (Smad) pathway, as INFβ activates the inhibitory Smads 6 and 7, which, in turn, suppress Smad1 phosphorylation and nuclear translocation, leading to suppressed BMP signaling transduction and consequently impaired osteoblast differentiation [39]. 

Figure 1 shows the tight interplay between the immune system and bone metabolism in SLE.

Along with inflammation, other factors affect bone mass in SLE. Glucocorticoids (GCs) are frequently used for treatment of flares and severe manifestations, leading to bone loss, and their use is associated to fragility fractures [22], as well as several chronic rheumatic diseases [40,41,42]. The greater rate of bone loss occurs within the first 3–6 months of GCs treatment, and a slower reduction lasts with chronic use. Both the high dose (>7.5 mg/day) and the length of treatment increase risk of fracture, particularly vertebral ones. Indeed, it has been previously shown that GCs have greater effects on trabecular bone than on cortical bone [16,22,33], owing to the more available surface of trabecular compartment upon which the cycle of resorption and formation can occur [23]. GCs act on the bone mass, both by reducing osteoblast function—through the enhancement of peroxisome proliferator activated receptor gamma receptor 2 expression, inhibition of Wnt/ß-catenin signalling pathway and reduction in the production of growth hormone and IGF, which are osteoblast stimulators—and by increasing osteoclast resorption, through an increase in RANKL and IL-6 production and reduction of OPG levels [22]. For these reasons, the American College of Rheumatology (ACR) Guideline for Glucocorticoid-Induced Osteoporosis has advised to perform routine BMD testing in patients starting long-term GCs therapy for autoimmune conditions, including SLE, within the first 6 months of GCs’ start, when rate of bone loss occurs more quickly [33]. The use of steroid-sparing medication is important to reduce disease activity but also to minimize dose and duration of GCs therapy in patients with frequent flares or chronic active disease [16]. Nevertheless, an interesting issue is how SLE therapy may positively affect bone mass. In particular, GCs might also have a favorable effect by reducing the adverse effect of systemic inflammation on bone, even if several studies about this topic have showed conflicting results. Bultink et al. stated that treatment with low dose prednisone (less than 7.5 mg daily) was not associated with bone loss. Thus, a proper use of GC therapy might prevent bone loss in patients with mild disease treated with lower dose of GCs [43].

Moreover, SLE patients often have low levels of vitamin D, which contribute to bone loss. The pathogenesis of hypovitaminosis D in SLE patients is multifactorial and it may be due to photosensitivity, with subsequent avoidance of sun exposure, use of sunscreens, renal failure, GCs use and probably hydroxychloroquine (HCQ) use, as HCQ is supposed to inhibit α1-enzyme hydroxylase which converts 25-hydroxyl-vitamin D in 1,25-dihydroxy vitamin D (1,25(OH)2D), as potential treatments [16,17]. Hormonal changes in patients with SLE may negatively affect bone mass: decreased level of dehydroepiandrosterone (DHEA) found in these patients are associated with low BMD [14,16,21].

Renal failure, which can develop as a consequence of the lupus nephritis occurring in up to 60% of the SLE patients, can contribute to the deterioration of bone mass because of the development of secondary hyperparathyroidism, increased osteoclastic bone resorption and reduced synthesis of 1,25(OH)2D [16,21].

Among serologic factors, the presence of anti-Ro has been associated with lower bone mass because anti-Ro positive SLE patients are usually advised against sun exposure, while presence of anti-Sm has been associated with higher BMD of the hip; the presence of antibodies against citrullinated protein (ACPA) or anti-carbamylated protein antibodies (antiCarP) in a small subgroup of SLE patients with erosive arthritis has been associated with significant bone loss, as it occurs in patients with rheumatoid arthritis [16,21].

Prevention and treatment of OP and fractures in SLE patients encompass general lifestyle measures, such as stopping smoking, limiting alcohol use, maintaining a normal body weight, prevention of falls and immobilization, regularly performing weight-bearing physical activity, an adequate calcium and vitamin D intake, and anti-osteoporotic drugs (anti-resorptive agents such as bisphosphonates and denosumab, and anabolic agents as teriparatide), suitable for patients with T-score below −2.5, with a previous fragility fracture or those receiving long-term GCs therapy [14,16,22,33].

## 3. SLE and Osteonecrosis

Osteonecrosis or avascular necrosis (AVN) is a serious and well-recognized complication in SLE patients, which may cause disability and affect quality of life because of pain and mobility limitation [44,45,46,47,48].

It is defined as necrosis of the bone marrow and trabecular bones due to inadequate blood supply, and may affect any bone, even more than one at different times; the femoral head and knee are the most commonly affected sites [48,49]. 

AVN has a multifactorial origin and is associated with several disorders and risk factors. The causes can be classified as either traumatic, related to an injury, such as fractures and dislocations, or atraumatic, consequent to GCs administration, alcohol use, Cushing disease, chronic renal failure, or smoking [49]. The underlying mechanisms that may cause an atraumatic AVN inadequate blood supply include increased bone marrow pressure and intravascular occlusion of subchondral vessels due to coagulation, fat emboli, thrombi, or abnormally shaped red blood cells. In addition, alcohol and drugs may exert direct toxicity toward bone cells [45]. 

### 3.1. Osteonecrosis in SLE Patients: Epidemiology, Risk Factors and Pathophysiology

Among rheumatic diseases, SLE is the most commonly associated with AVN [46,48]. AVN was first described as an SLE complication in the 1960s and has been frequently reported in these patients [47,48]: prevalence of symptomatic AVN is reported to be 0.8–33%, this difference being due to variability in diagnostic tools and study methodology, but it can have a clinically silent course. The prevalence of asymptomatic AVN is reported to be 29–45% [24,47], but its incidence has gradually decreased over the past two decades, probably owing to the improvement in SLE management and lowering use of GCs [44,47]. 

In SLE, AVN mainly affects weight-bearing joints and femoral head is the most commonly affected site; bilateral hip involvement has been reported in up to 90% of SLE patients with AVN [44,45,47,50]. Multifocal osteonecrosis is defined as involvement of at least three sites, and it is usually associated with GCs; it is an uncommon condition in SLE, as described in some case reports [48,51]. 

Several AVN risk factors in SLE are shown in Table 3.

GCs therapy has been reported as one of the most important risk factors for AVN development in SLE [44,45,46,47,48,51]. The average daily dose, the cumulative dose of GCs and Cushing syndrome were reported to be associated with AVN [44,45,46,48]. In particular, a daily prednisone dose of 20–39 mg increased the risk only when administered for more than one month, while a daily dose above 40 mg is associated with increased AVN risk even when administered for a shorter time; therefore, high-dose GCs treatment >40 mg/day can be considered as the threshold that predisposes to development of AVN [44,46,54]. The association between GCs treatment and AVN is not fully understood, but several mechanisms have been proposed. Chronic GCs use causes differentiation of pluripotent mesenchymal cells to adipocytes and increases intracellular fat accumulation, promoting intraosseous adipocyte hypertrophy and fat conversion of red marrow; consequently, bone marrow pressure increases and impairs bone perfusion, leading to bone ischaemia [44,45,47,48]. Moreover, GCs may affect lipid metabolism, so that fat microemboli may occlude subchondral vessels [45], and may induce vasoconstriction of bone arteries by means of vasoactive agents as endothelin-1 and bradykinin [46,47]. 

Nevertheless, it is possible to observe AVN in SLE patients without a history of GCs treatment and most SLE patients who receive GCs do not develop AVN in the course of the disease; further, increased incidence of AVN in SLE compared to the general population, to other autoimmune diseases, and to other diseases treated with high GCs doses suggests that additional SLE-specific risk factors may be involved [44,45,46,47,48,51]. Vasculopathy, abnormal endothelial function, dyslipidemia and fat embolism have been reported to be associated with AVN in SLE, despite some controversial evidence [46,47]. 

AVN may be an osteoarticular manifestation in primary antiphospholipid syndrome (APS), sometimes as a presenting feature; some of these patients may not have a history of GCs intake, suggesting a role for antiphospholipid antibodies (aPL) in the pathophysiology of AVN [44,45,46,47,52]. In addition, aPL may cause endothelial damage and act as a prothrombotic factor leading to microvascular thrombosis in terminal-end arteries, bone ischaemia and tissue death [44,45,46,55]. Particularly, AVN has been reported as associated with anticardiolipin (aCL) IgG subtype [47] and mainly aCL IgM subtype [24,45]. 

The relationship between Raynaud’s phenomenon, livedo reticularis and vasculitis with AVN has been investigated with conflicting results [44,45,46].

It has been reported that African-American SLE patients have a higher risk of developing AVN, probably due to more severe disease requiring higher doses of GCs [44,47]. Recently, a possible genetic predisposition has been identified: variant alleles of the Apolipoprotein L1 (*APOL1*) gene, which are involved in innate immune responses and more expressed in African-American patients, are related with an increased risk of progressive kidney disease and atherosclerotic disease, and have been reported to be more prevalent in African-American SLE patients with AVN, supporting a possible role in AVN physiopathology [44,47,56].

Disease activity is a significant and independent risk factor for development of AVN [24,44,45,46,47,53]. SLE patients with AVN present a high disease activity (as defined by an SLEDAI score > 8) than patients without AVN [45,53]. Clinical manifestations of active SLE (neuropsychiatric manifestations, renal disease, serositis, cytopenias) and younger age of disease onset are associated with AVN [24,44,45,46]. Other clinical features, such as arthritis, oral ulcer, malar rash, GC-induced OP, gastrointestinal involvement, have been associated with AVN [44,45,46,47,53]. 

Several drugs used in SLE treatment may affect the risk of development of AVN. Use of antimalarial drugs may protect against AVN development, due to their antithrombotic and lipid-lowering properties, even if available data are conflicting [45,46,47,53]. Immunosuppressive treatment with cytotoxic agents, in particular cyclophosphamide, may exert a direct negative effect on osteoblasts and osteoclasts and represent a risk factor for AVN; moreover, these drugs are used in patients with severe clinical features and high disease activity, which are themselves risk factors of AVN [45,47,53].

### 3.2. Osteonecrosis Imaging

Since AVN may be clinically asymptomatic, the identification of known risk factors and imaging play an important role for identifying this clinical manifestation [45,47,48,53]. Further, patients should be educated on presenting symptoms, namely, progressive or sudden joint pain and impaired range of motion [45]. 

Several imaging techniques allow AVN detection, even in early and asymptomatic phases. Magnetic resonance imaging (MRI) plays a key role in the diagnosis of AVN, due to higher sensitivity compared to traditional radiography and higher specificity compared to bone scintigraphy [48,50]. MRI early findings consist of a central necrotic area with preserved fat signal bordered by an irregular sclerotic and reactive rim, resembling a “double line sign” on T2-weighted sequences. Later, fibrotic tissue may appear as a low intensity signal in both T1-weighted and T2-weighted sequences. Other findings consist of bone marrow oedema, synovial effusion, and secondary osteoarthritic changes [57]. Moreover, SLE patients show a higher amount of bone marrow fat and a higher degree of conversion of red marrow into fatty bone marrow than healthy controls [48]. Radiography may detect late changes, such as bone fragmentation, joint space narrowing, and secondary osteoarthritis [57]. Technetium-99m methylene diphosphonate bone scintigraphy highlights increased osteoblast activity and hyperaemia of bone, and may be used for the early detection of AVN, despite its low specificity. Early AVN generally appears as an area of reduced bone tracer uptake, but within days increased radiotracer uptake may be observed due to osteoblastic activity at the margins of the necrotic segment [50,58]. Bone scintigraphy is useful in assessment of multifocal AVN [57]. 

### 3.3. Osteonecrosis Management in SLE

AVN management encompasses several measures in SLE patients. A proper use of GCs, and the control of disease activity score and other risk factors may reduce AVN development [45,47,53]. Pharmacological treatment of AVN in SLE patients is similar to that of AVN in the general population, which may range from analgesics and physical therapy for early stages to arthroplasty for more severe stages [48]. Bisphosphonates used as a prophylactic treatment against GC-induced OP have been associated with a significantly lower AVN prevalence among SLE patients; in addition, these drugs may relieve AVN symptoms, delaying disease progression and surgery [47].

## 4. SLE and Osteomyelitis

Infections strongly affect morbidity and mortality in SLE [57,59]. Herrinton et al. reported a six- to seven-fold greater risk of serious infection than the general population [60], mainly due to impaired function of immune system and immunosuppressive treatment [59,61]. Infections may be life-threatening and represent one of the leading causes of death in SLE [60,62,63], accounting for 25% to 50% of overall mortality [63]. Respiratory and urinary tracts and soft tissue represent common infection sites in SLE [59]. 

### 4.1. Osteomyelitis in SLE Patients: Epidemiology, Risk factors and Pathophysiology

Osteomyelitis (OM) is a rare but potentially life-threatening complication in SLE patients [57,59]. OM is a bone infection caused by various microorganisms, which may involve trabecular and cortical bone, bone marrow and periosteum [64]. It may be classified according to patho-physiologic mechanisms: hematogenous OM due to bacteraemia, OM caused by local spreading from a contiguous focus of infection (from trauma, surgery, prosthetic material or soft tissue) and OM due to vascular insufficiency [64,65]. 

In addition to environmental factors, genetic factors may contribute to OM pathogenesis. Despite appropriate treatments, up to 30% of OM cases become chronic, resulting in serious morbidity and disability and remarkable economic burden [66]; a genetic predisposing substrate may partly explain these results. Recently, Xie et al. analyzed several single-nucleotide polymorphisms (SNPs) of DNA sequences linked to the susceptibility of OM and they reported that the SNPs in *IL1B*, *IL6*, *IL4*, *IL10*, *IL12B*, *IL1A*, *IFNG*, *TNF*, *PTGS2* (prostaglandin-endoperoxide synthase 2), *CTSG* (cathepsin G), *VDR* (vitamin D receptor), *MMP1* (matrix metalloproteinase 1), *PLAT* (plasminogen activator, tissue type), and *BAX* (Bcl-2 associated X) increased the risk of bacterial OM, whereas those in *IL1RN* (IL-1 receptor antagonist) and *TLR2* (toll-like receptor 2) could protect against OM [66]. These OM-related genes encode for cytokines, enzymes and proteins involved in inflammatory pathways and in host defense against pathogenic microorganisms, suggesting that inflammation plays a pivotal role in OM development.

OM is more frequent in SLE patients than in the general population [57,59], moreover occurring at a younger age [57]. Several studies and case reports described OM in SLE, due to different microorganisms [50,59,61,67,68,69,70] and affecting mainly bones of the lower limbs, in particular tibia and femur [57,59]. 

Different conditions may be associated with higher risk of OM development in SLE, particularly impaired immune system, GCs and immunosuppressive treatment, high disease activity, underlying bone and joint pathology such as bone infarcts or prior bone fracture, old age (>60 years) and pediatric age, male gender, and malignancy [59,61,63,67,68,69,70]. 

Impaired immune system in SLE can increase risk of infections through the decreased phagocytosis, reduced production of IL-8 and IL-12, complement deficiency, impaired chemotaxis and membrane recognition that have been described in these patients [63,67]. 

GCs predisposing OM may be related to dose and duration of treatment [59,61,63]. As reported by Huang, a prednisone equivalent dose of more than 7.5 mg for more than 180 days was associated with increased risk of major infection and OM in SLE [59]. Thus, GCs may increase the risk of opportunistic infections, including OM [59,61]. Immunosuppressive treatment may contribute to increased risk of infection and OM [59]; among immunosuppressive drugs, cyclophosphamide could deeply reduce immune system function, inducing leukopenia [63]. 

Malignancy and its treatments may be considered as an independent risk factor for hematogenous or contiguous OM in immunocompromised SLE patients [59]. 

Several bone diseases may increase the risk of OM development in SLE. AVN, a common complication in SLE as reported above, may produce a bone environment that facilitates persistent bone infection, due to impaired blood supply [68]. Fractures may predispose to direct inoculation of infection into the bone [59]; stress fractures, frequently seen in SLE patients, may be associated with AVN and later complicated by OM [67]. 

### 4.2. Clinical Manifestations of Osteomyelitis in SLE Patients

Clinical manifestations of SLE flare and OM, as fever and musculoskeletal pain, may be similar, potentially leading to delayed diagnosis and treatment of OM [59,63], and GCs treatment may mask infection symptoms [50]. Moreover, differential diagnosis may encompass other diseases, such as septic arthritis and fractures [61]. SLE patients with OM usually report subacute or chronic onset of pain at site of bone involvement, fever and chills may be present and soft tissue erythema, swelling and cutaneous fistulae may be observed [65]. OM should be suspected in case of new-onset or worsening musculoskeletal pain, especially if associated with fever and/or bacteremia, traumatic injury, septic arthritis, chronic skin ulcers near bony structures, soft tissue infection adjacent to orthopedic implants [57].

Laboratory tests and imaging techniques are fundamental to identify OM, mainly to distinguish between OM and an SLE flare, which requires antithetic treatment [63].

### 4.3. Osteomyelitis Imaging

Imaging plays an important role in the early detection and assessment of OM. Radiographic findings of OM may not be obvious or specific within the first two weeks following bone infection in up to 80% of patients; thereafter, loss of trabecular architecture and cortical bone, periosteal thickening, osteopenia and soft tissue swelling may be detected [57,61]. In chronic phases (more than 6 weeks), Rx findings may be represented by sequestration of bone (necrotic bone separated from healthy bone), a thick sheath of periosteal new bone surrounding sequestrum areas and reactive bone sclerosis [57]. MRI represents the gold standard for OM assessment [61], which allows early detection (less than 2 weeks after bone infection) of bone marrow oedema, joint and soft tissue inflammation, and assessment of extent of bone destruction [57]. T1-weighted sequence with fat-suppression after gadolinium administration shows areas of necrotic, non-enhancing bone and soft tissue with peripheral contrast enhancement [71]. CT and nuclear imaging may be considered as an important diagnostic tools when MRI is contraindicated. CT shows bone changes, such as sequestrum areas, cortical and trabecular changes and periosteal reaction, better than MRI, but it cannot detect bone marrow oedema [57,61]. A three-phase bone scintigraphy with technetium-99m-methylene diphosphonate identifies focally increased bone uptake, but it is characterized by low specificity [57,71]. Radio-labelled leukocyte scan is characterized by greater specificity, although it cannot distinguish between OM and soft tissue infection [57]. Leukocytosis, increased level of C-reactive protein and procalcitonin may suggest OM diagnosis [59,61,63]; blood, urine, and stool cultures should be performed in case of suspected OM [67]. Conversely, disease activity markers, such as low level of complement, high anti-ds-DNA antibody, leukopenia, in patients with fever and other SLE-related manifestations, such as arthritis and skin rash, may indicate an SLE flare [63].

### 4.4. Osteomyelitis Management in SLE

OM management may involve several specialists (rheumatologist, infectious disease specialist, orthopedist, radiologist), due to its complexity [68]. Treatment encompasses long-term antimicrobial therapy, immobilization and surgical debridement, such as bacteria adhere to biofilm in necrotic avascular tissue, being poorly susceptible to antibiotics, without debridement [61,67,68]. Before starting empirical treatment, tissue specimens should be collected for Gram strain, culture, and antibiogram, and histopathology, in order to initiate proper antibiotic therapy; this therapy is usually administered intravenously and lasts four to eight weeks, even longer in chronic OM [72].

## 5. Conclusions

SLE is a chronic autoimmune disease with a broad spectrum of clinical signs. Overall musculoskeletal manifestations are highly frequent, reported in up to 90% of patients [57]. Particularly, bone involvement is one of the most common and disabling manifestations. An increased incidence of OP, AVN and OM has been observed in SLE patients than the general population. Early detection of these potential complications is an actual issue, since they can affect patients’ quality of life, morbidity and mortality, and healthcare burden. Moreover, some signs of these complications may be mistaken with clinical manifestations of SLE, in particular with flares, requiring a different management: particularly, a lupus flare requires immunosuppressive drugs, while treatment of infectious manifestations such as OM relays on antimicrobial agents. Thus, all clinical signs, laboratory tools and imaging allows a proper differential diagnosis and management.

## Figures and Tables

**Figure 1 ijms-23-05804-f001:**
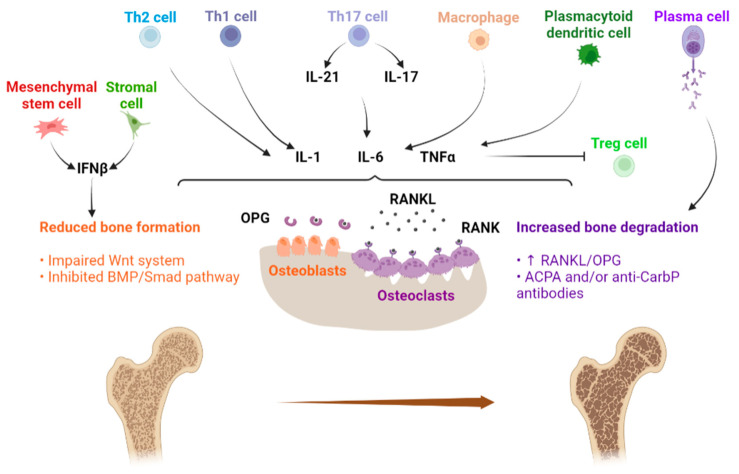
Osteoimmunology: the tight interplay between immune system and bone metabolism in SLE. Immune cells interplay affecting bone metabolism, mainly by means of a proinflammatory cytokine network that impairs the balance between osteoclast and osteoblast function. ACPA: anti-citrullinated protein antibody. Anti-CarbP: anti-carbamylated protein. BMP: bone morphogenetic protein. IFNβ: interferon β. IL-1: interleukin 1. IL-6: interleukin 6. IL-17: interleukin 17. OPG: osteoprotegerin. RANK: receptor activator of nuclear factor-kB. RANKL: receptor activator of nuclear factor-kB ligand. SLE: systemic lupus erythematous. Smad: small mother against decapentaplegic. Th1 cells: T helper 1 cells. Th2 cells: T helper 2 cells. Th17 cells: T helper 17 cells. TNFα: tumor necrosis factor α. Treg cells: T regulatory cells. Wnt: wingless-related integration site. Original figure.

**Table 1 ijms-23-05804-t001:** Osteoporosis risk factors in SLE.

Traditional Independent Risk Factors	Disease-Related Risk Factors
Low body mass index [12,13,14,16]	Systemic inflammation [14,16,17,18,19]
Old age [11,12,13,14,16]	Lupus nephritis [14,16,17,21]
Female gender [11,12,13,14,16]	High levels of oxidized LDL [14,16,17]
Postmenopausal status [11,12,13,14,16]	Hormonal abnormalities [14,16,17,21]
Smoking [13,14,16]	Presence of anti-Ro, ACPA and/or antiCarP antibodies [14,15,16,21]
Alcohol abuse [13,14,16]	Medication [12,13,14,16,17,19,22,23,24]
Low level of vitamin D [12,13,14,16]	Immobility [14,15,16,17]
Low intake of calcium [12,13,14,16]	
Low physical activity [12,13,14,16]	
Hypogonadism [12,13] Family history of osteoporosis [11,12,13,14,16]	

**Table 2 ijms-23-05804-t002:** Osteoporotic fracture risk factors in SLE.

Independent Risk Factors	Disease-Related Risk Factors	Medication-Induced Adverse Effects	Bone Impairment
Old age [11,12,13,14,16]	Disease duration [14,16,17,18]	Glucocorticoids [14,16,17,22,23,33]	Low BMD [11,12,13,14,16,17,19]
Female gender [11,12,14,16]	Seizures [16,17,18]	Antiepileptic drugs [16]	Low TBS [19,28,31]
Smoking [13,14,16]	History of stroke [16,18]	Anticoagulants [16]	Previous fragility fractures [13,16,18]
Alcohol abuse [13,16]	Renal failure [14,16,21]	Cyclophosphamide [16]	
Postmenopausal status [11,12,14,16]	Presence of lupus anticoagulant [14,16]		
Frailty [16,32]			
Obesity [14,16]			
Low level of vitamin D [12,13,14,16]			

**Table 3 ijms-23-05804-t003:** Avascular necrosis risk factors in SLE.

Independent Risk Factors		Disease-Related Factors
Traumatic-Associated Risk Factors	Atraumatic-Associated Risk Factors	
Fractures [49,52]	Hyperlipidemia [44,45,49]	High disease activity [24,44,45,46,47,53]
Dislocation or fracture- dislocation [48,49]	Chronic renal failure/ hemodialysis [44,47,48,49]	Medication-induced adverse effects [44,45,46,47,48,51,53,54]
Radiation [49]	Smoking [44,47,48,49]	Antiphospholipid syndrome [24,44,45,46,47,52,55]
Other comorbidities (i.e., hematologic diseases) [45,48,49]	Alcohol use [44,45,48,49]	Vasculopathy and abnormal endothelial function [44,45,46,47]
	Organ transplantation [49]	Clinical features [24,44,45,46,53]
	Hyperuricemia/gout [49]	Genetic factors (*APOL1* variant alleles) [44,47,56]
	HIV [49]	Younger age of disease onset [24,44,45,46]
	Intravascular coagulation [44,49]	
	Thrombophlebitis [49]	
	Cushing disease [44,49]	
	Drugs (GCs) [45,47,48,49,51]	
	Pancreatitis [49]	
	Pregnancy [49]	

## Data Availability

The authors declare that the data supporting the findings of this study are available within the paper.

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
