# Peer review of "Bone Involvement in Systemic Lupus Erythematosus"

_ijms, 2022, doi:10.3390/ijms23105804_

Round 1

Reviewer 1 Report

It is a well-documented paper, congratulations.

Author Response

No revision requested

Reviewer 2 Report

good  article, minor revision is recommended I recommend some modifications and propositions: 1. please add the databases and the period of the study; there is an article from 1999 and other from 2022. If the studied period is from 1999 to 2022, the authors should add all the articles to their review. 2. Did the authors have the authorization for the Figure 1? or it's an original figure? 3. Lot of abbreviations, I propose to make a part for all the abbreviations in the text 4. I didn't find the citations of REF: 69 and 70, please clarify

Author Response

  1. The database used and the period of the study have been added. All selected published papers were included in the original version of the manuscript
  2. Figure 1 is an original figure, as specified in the legend of figure

  3. The abbreviations have been added at the end of the text, after “Conclusions” paragraph

  4. References 69 and 70 have been cited in lines 394-395 and 397-401

Reviewer 3 Report

The paper is a review of bone manifestations in SLE patients. Although limited by its narrative design, the paper is interesting and current. I have only few minor comments for the authors:

  • Page 3, lines 83-87. Please add the HRs for sex and age.
  • Page 5. In figure 1 the authors reported the role of IL-21 in bone metabolism, however they reported no data in Section 2.3. Please consider to provide some insight.
  • Page 6, lines 224-226. Please provide appropriate reference for the statement.

Author Response

  1. The HRs have been added, as required (lines 93-95)
  2. Data about the role of IL-21 have been added (lines 162-163)

  3. Appropriate reference has been added (lines 232-234, reference 16)